# Dapsone-Associated Anemia in Heart Transplant Recipients with Normal Glucose-6-Phosphate Dehydrogenase Activity

**DOI:** 10.3390/jcm11216378

**Published:** 2022-10-28

**Authors:** Kevin W. Lor, Evan P. Kransdorf, Jignesh K. Patel, David H. Chang, Jon A. Kobashigawa, Michelle M. Kittleson

**Affiliations:** Department of Cardiology, Smidt Heart Institute, Cedars Sinai Medical Center, 127 S. San Vicente Blvd., Los Angeles, CA 90048, USA

**Keywords:** dapsone, transplant, G6PD, anemia

## Abstract

Dapsone is considered an alternative for pneumocystis jirovecii pneumonia (PJP) prophylaxis in sulfa-allergic or -intolerant transplant patients with normal glucose-6-phosphate dehydrogenase (G6PD) activity. Despite normal G6PD activity, anemia can still occur while on dapsone therapy. We retrospectively reviewed heart transplant patients transplanted at our center between January 2016 and June 2018 and identified those taking dapsone prophylaxis. There were 252 heart transplant recipients at our center between January 2016 and June 2018. 36 patients received dapsone prophylaxis. All had normal G6PD activity assessed prior to dapsone initiation. 8 (22%) patients developed significant anemia attributed to dapsone: 2 were hospitalized for anemia, 1 of whom required blood transfusion. These patients had a median reduction in hemoglobin of 2.1 g/dL from baseline prior to dapsone initiation. Overt evidence of hemolysis was present in six patients. Once dapsone was discontinued, Hgb increased by at least 2 g/dL in a median of 30 days. Anemia from dapsone may occur in a significant proportion of patients despite normal G6PD activity and resulting in significant morbidity. Careful monitoring of transplant recipients on dapsone prophylaxis is warranted, as well as consideration of alternative agents.

## 1. Introduction

*Pneumocystis jirovecii* is a ubiquitous fungus that can cause pneumonia in up to 15% of transplant recipients in the absence of appropriate antimicrobial prophylaxis. The drug of choice for *Pneumocystis jirovecii* pneumonia (PJP) prophylaxis is trimethoprim-sulfamethoxazole for a duration of 6 to 12 months, but its utility can be limited by allergic reactions present in approximately 3% of the population [1,2].

In patients with trimethoprim-sulfamethoxazole intolerance or allergy, dapsone is often used for PJP prophylaxis. Dapsone inhibits bacterial and protozoan synthesis of dihydrofolic acid [3]. Before initiation, glucose-6-phosphate dehydrogenase (G6PD) function should be checked as a deficiency increases the risk of hemolytic anemia and methemoglobinemia [1,4].

Dapsone may also cause dose-dependent anemia [5,6,7], though the incidence of anemia in heart transplant patients with normal G6PD activity on dapsone is not well characterized [3,8,9,10,11,12]. The purpose of this study was to report the incidence and characterization of dapsone-associated anemia with normal G6PD function in heart transplant recipients at our center.

## 2. Materials and Methods

The study was approved by our Institutional Review Board at Cedars Sinai. We reviewed patients transplanted at our medical center between January 2016 and June 2018 to identify those who received dapsone prophylaxis. The electronic medical record was used to extract clinical data including demographic information, clinical characteristics, and laboratory values. All patients had normal G6PD activity prior to initiating dapsone. G6PD activity was a send-out lab to Quest Diagnostics Nichols Institute with normal values 7.0–20.5 units/gram of hemoglobin. All patients were maintained on a calcineurin inhibitor, mycophenolate mofetil, and prednisone. While patients were admitted, basic metabolic panel (BMP) and complete blood count (CBC) labs were drawn daily. Outpatient BMP and CBC labs were drawn twice weekly until month 1 after transplant, weekly until month 2, every other week until month 3, monthly until month 6, then every 3 months until the first year, unless patients were re-admitted. Additional labs were ordered only when deemed necessary.

Hemolytic anemia was defined by a decreased hemoglobin with any of the following: presence of spherocytes, schistocytes, or bite cells on peripheral blood smear, symptoms such as brown urine or jaundice; and laboratory findings such as elevated lactate dehydrogenase (>220 U/L), elevated indirect bilirubin (>1 mg/dL), decreased haptoglobin (<36 mg/dL), elevated reticulocyte percentage (>2%) and macrocytosis (>100 fL) [13]. As reticulocytes are larger than mature erythrocytes, the presence of macrocytosis can be a surrogate for reticulocytosis, after ruling out other causes of macrocytosis like vitamin B-12 or folate deficiencies [14].

Statistics were calculated using the chi-squared test and paired *t*-test.

## 3. Results

Between January 2016 and June 2018, 252 patients underwent heart transplantation at our center. Of these, 36 patients (14%) received dapsone, most commonly 100 mg daily, for PJP prophylaxis. G6PD activity was assessed prior to initiation of dapsone in all patients and was normal. Dapsone was initiated at a median of 12.5 days (range 2 to 157 days) post heart transplantation. The reasons for the use of dapsone included: documented sulfa allergy causing a rash or anaphylaxis (21 patients), acute kidney injury (8 patients), leukopenia (4 patients), hyperkalemia (2 patients) and elevated alkaline phosphatase (1 patient).

Of the 36 patients who received dapsone for PJP prophylaxis, 8 (22%) developed anemia that resolved with discontinuation of dapsone. There was no difference in age, gender, prior durable mechanical circulatory support device, antithymocyte induction use, ethnicity, dapsone dosage, and reason for dapsone initiation between those patients with anemia and without anemia (Table 1). Of these 8 patients with anemia, 2 were hospitalized for anemia and one required blood transfusion.

Six of the patients met at least one laboratory or clinical criteria for hemolytic anemia, but none of the patients had complete hemolysis labs and four had peripheral blood smears (Table 2).

Dapsone was discontinued after a median of 52 days (range 21–90 days) from initiation and replaced with atovaquone (Table 3). At least 30 days after dapsone discontinuation, at a median of 40 days (range 30–55 days), the hemoglobin increased 39% from nadir, a median of 3 g/dL (range 1.9–4.9 g/dL). Figure 1 depicts the degree of anemia while on dapsone therapy and hemoglobin recovery at least 30 days after dapsone was discontinued. The time course of resolution was that hemoglobin increased by 2 g/dL from nadir occurred after a median of 30 days (range 15–54 days). Seven patients had macrocytosis. MCV decreased to normal, under 100 fL, at a median of 42 days (range 30–65 days) post dapsone discontinuation. One patient was restarted on dapsone 51 days after discontinuation, which resulted in recurrent anemia, hemoglobin 10.8 g/dL off dapsone to a nadir of 8.2 g/dL after 73 days on dapsone for which dapsone was stopped a second time. Another patient reported only “dark brown urine” per chart description without anemia after one week of dapsone therapy, which resolved after discontinuation. Upon rechallenge six days later, this patient again reported “dark urine” and developed anemia over 2 weeks.

Patients were scored on the Naranjo Adverse Drug Reaction (ADR) Probability Scale which provides a standardized assessment of causality for adverse drug reactions [15]. The reaction is considered definite if the score is 9 or higher, probable if 5 to 8. Two patients scored +11, indicating a definite causal relationship and the rest scored +8, indicating a probable causal relationship between dapsone use and anemia (Table 4).

## 4. Discussion

Despite normal G6PD activity, 22% of heart transplant recipients receiving dapsone for PJP prophylaxis developed anemia with at least a probable causal association by the Naranjo Adverse Drug Reaction Probability scale. This anemia resulted in significant morbidity, with 2 patients requiring hospitalization and one requiring blood transfusion.

To our knowledge, this is the first reported case series of dapsone-associated anemia in heart transplant recipients with normal G6PD activity though this observation has been made in other solid organ transplant (SOT) recipients. Dapsone-related anemia has been reported in the setting of normal G6PD activity resulting in dapsone discontinuation in 46% of kidney transplant recipients [12] and 23% of lung transplant recipients [11]. One small study observed dapsone-related anemia in kidney, lung, liver, and heart transplant recipients though G6PD activity was not documented in all patients [9]. Dapsone-related anemia has also been observed in non-SOT patients, up to 87% in stem cell transplant recipients with normal G6PD activity [10], 4% in patients with HIV [16], and 25% of patients with leprosy [3]. The incidence of dapsone-related anemia of 22% in our study is comparable to that observed in lung transplant recipients but lower than that seen in kidney or stem cell transplant recipients, which may be related to comorbidities or duration or dosage of dapsone used.

Dapsone is absorbed almost completely with 80–100% bioavailability. Only 20% of dapsone is renally eliminated unchanged while 70–85% of dapsone metabolites are renally eliminated [5]. All of the patients with dapsone-associated anemia did not have any renal dysfunction to explain any dapsone or dapsone metabolite accumulation.

One putative mechanism for dapsone-induced hemolytic anemia in patients with normal G6PD activity is the accumulation of the toxic metabolite dapsone hydroxylamine which forms free radicals in erythrocytes [17,18]. This accumulation may be modified based on genetic polymorphisms. Dapsone undergoes metabolism via acetylation by N-acetyltransferase to an inactive metabolite (monoacetyldapsone) and oxidation primarily by the cytochrome P450 2E1, but also 3A4 and 2C9 [5,19,20], to its toxic metabolite dapsone hydroxylamine. It is possible that patients who are both slow acetylators and fast oxidizers are more prone to dapsone-induced anemia as the predominant metabolite would be shifted from the inactive metabolite to the toxic one [17,21,22]. The role of pharmacogenomics on the metabolism and toxicity of dapsone is an important area of future study.

This study has several limitations. First, the small size precludes clear assessment of causality. The retrospective nature also prevented clear assessment of hemolysis as relevant laboratory assessments for the diagnosis of hemolysis were not performed in all patients. Patients were managed as an outpatient which contributed to delays in recognition of anemia or extended duration of dapsone. Furthermore, in some patients the anemia was gradual, whereas in others, the anemia was pronounced. However, once dapsone was discontinued, the anemia gradually resolved regardless of the onset of anemia. Although only 2 of 6 patients exhibited low haptoglobin, the haptoglobin levels could have been falsely elevated by corticosteroids which are given post heart transplantation, thereby overlooking the diagnosis of hemolysis [23]. Nonetheless, these real-world observations provide important insight for clinicians managing solid organ transplant patients with worsening anemia while receiving dapsone prophylaxis.

## 5. Conclusions

Dapsone is used regularly for PJP prophylaxis in patients who are allergic or intolerant to sulfonamide antibiotics. Despite normal G6PD function, hemolytic anemia can still occur in patients on dapsone prophylaxis leading to potential hospitalizations and blood transfusions. Careful monitoring is necessary and alternative agents for PJP prophylaxis should be considered.

## Figures and Tables

**Figure 1 jcm-11-06378-f001:**
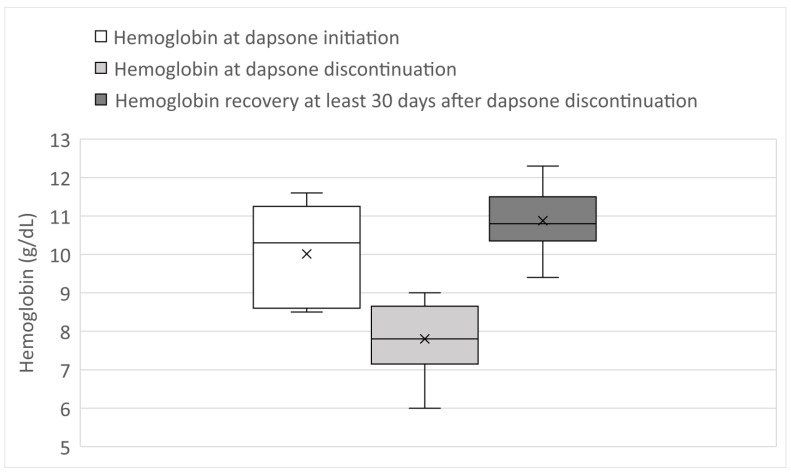
Hemoglobin levels at dapsone initiation, at dapsone discontinuation and at least 30 days after dapsone discontinuation with interquartile ranges, ranges, medians, and means (marked by x). *p* = 0.0018 comparing hemoglobin at dapsone initiation to hemoglobin at dapsone discontinuation. *p* < 0.0001 comparing hemoglobin at dapsone discontinuation to hemoglobin recovery.

**Table 1 jcm-11-06378-t001:** Demographics.

	Hemolytic Anemia (n = 8)	No Hemolytic Anemia (n = 28)	*p*-Value
Mean age +/− SD	54.9 +/− 14.2	57.0 +/− 10.3	0.64
Female (%)	4 (50)	5 (18)	0.07
Prior durable MCS device (%)	2 (25)	10 (36)	0.57
ATG induction (%)	8 (100)	22 (79)	0.16
Ethnicity White African American Hispanic Asian	4 (50) 0 (0) 3 (37) 1 (13)	18 (64) 7 (25) 2 (7) 1 (4)	0.06
Daily dapsone dose 100 mg 50 mg	7 (87) 1 (13)	25 (89) 3 (11)	0.88
Reason for dapsone initiation Sulfa intolerance Kidney injury Leukopenia Hyperkalemia Elevated alkaline phosphatase	6 (75) 0 (0) 1 (13) 0 (0) 1 (13)	15 (54) 8 (29) 3 (11) 2 (7) 0 (0)	0.21

**Table 2 jcm-11-06378-t002:** Evidence of hemolysis.

Patient Number	G6PD Level	Baseline Hgb (g/dL)	Hgb Nadir (g/dL)	Hgb (g/dL) at Least 30 Days after Dapsone Discontinuation	Trans-Fusion (no. of PRBC Units)	Hapto-Globin (Normal: 36–195 mg/dL)	LDH (Normal: 125–220 U/L)	Reticul-ocyte % (Normal: 0.5–2%)	Schisto-Cytes
1	Normal	8.5	7.5	9.4	0	85	244	10.7	no
2	Normal	8.5	7.4	12.3	0	<8	---	3.5	---
3	Normal	11.4	6	10.6	0	138	468	7.4	yes
4	Normal	8.7	6.9	10.8	0	215	266	3.6	no
5	Normal	11.1	8.3	11.3	0	<8	---	11.2	---
6	Normal	9.2/10.8	8.6/9	10.5/11.1	0	---	---	---	---
7	Normal	10.3	7.8	10.2	2	208	248	6.3	no
8	Normal	11.6	8.7	11.7	0	---	---	---	---

**Table 3 jcm-11-06378-t003:** Dapsone duration and recognition of anemia.

Patient Number	Daily Dose (mg)	Days from Date of Transplant to Dapsone Initiation	Days from Dapsone Initiation to Onset of First Hgb Drop	Days from Dapsone Initiation to Hgb Nadir	Days from Date of Transplant to Dapsone Discontinuation	Days to Discontinuation of Dapsone after Initial Hgb Drop
1	100	3	15	22	30	15
2	100	5	11	23	47	31
3	100	7	7	50	57	43
4	100	6	11	21	27	10
5	100	63	46	46	116	7
6 *	100	25/166	90/45	90/101	115/267	0/56
7	50	18	13	76	94	63
8	100	58	59	59	117	0

* Patient 6 was rechallenged with dapsone and exhibited anemia when dapsone was restarted that again resolved when it was discontinued.

**Table 4 jcm-11-06378-t004:** The Naranjo probability scale to determine the likelihood of causation of dapsone and anemia. Two patients scored definite causal relationship while six patients scored probable causal relationship. Adapted from Naranjo et al. [15].

Patient	1	2	3	4	5	6	7	8
Has this adverse event been documented before? (+1 Y, 0 N)	+1	+1	+1	+1	+1	+1	+1	+1
Did the adverse reaction occur after suspected drug was given? (+2 Y, −1 N)	+2	+2	+2	+2	+2	+2	+2	+2
Did the adverse reaction resolve after cessation of drug or was it reversible? (+1 Y, 0 N)	+1	+1	+1	+1	+1	+1	+1	+1
Did the adverse reaction recur after re-challenge with suspected drug? (+2 Y, −1 N)	+2	0	0	0	0	+2	0	0
Have other causes been ruled out? (−1 Y, +2 N) *	+2	+2	+2	+2	+2	+2	+2	+2
When an alternative was given, did the reaction occur? (−1 Y, +1 N)	+1	+1	+1	+1	+1	+1	+1	+1
Was there any determination of toxic drug levels in the blood or other fluids? (+1 Y, 0 N)	0	0	0	0	0	0	0	0
Did changing the dose change the severity of the reaction? (+1 Y, 0 N)	0	0	0	0	0	0	0	0
When the patient was given the drug or alternative previously, did they experience a reaction? (+1 Y, 0 N)	+1	0	0	0	0	+1	0	0
Was there any objective evidence to verify the adverse effect? (+1 Y, 0 N)	+1	+1	+1	+1	+1	+1	+1	+1
Total (> +9: definite, +5–8 probable, possible +1–4, doubtful < +1	+11	+8	+8	+8	+8	+11	+8	+8

* +2 chosen because the anemia recovered after stopping dapsone and the onset of anemia was 30 days or later after transplant, so post-surgical anemia unlikely. In addition, the anemia occurred and resolved with initiation and discontinuation of dapsone regardless of maintenance on all other potential contributing medications including mycophenolate mofetil and valganciclovir.

## Data Availability

Data available upon request.

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
