# Peer review of "Dapsone-Associated Anemia in Heart Transplant Recipients with Normal Glucose-6-Phosphate Dehydrogenase Activity"

_jcm, 2022, doi:10.3390/jcm11216378_

Round 1

Reviewer 1 Report

Dear authors, 

I have read your manuscript with great interest.

It is a well-written manuscript covering an important topic in the post-transplant management. Results are clearly presented and the discussion covers and solves all possible questions.

I would only suggest to extend the study period if possible.

Author Response

Point 1: It is a well-written manuscript covering an important topic in the post-transplant management. Results are clearly presented and the discussion covers and solves all possible questions

Response 1: Thank you!

Point 2:  I would only suggest to extend the study period if possible

Response 2: Based on our findings, we have decreased our use of dapsone for PJP prophylaxis and are unable to extend the study period

Reviewer 2 Report

The authors report dapsone associated anemia in a cohort of heart transplanted patients with normal G6PD activity. The report is well written and interesting. Anemia is a well known side effect of dapsone but is not presented in heart transplantation previously. The main limitation is the limited number of patients but as anemia was seen in 22% of them, this is probably sufficient to say that this is a potential complication of dapsone in heart transplant.

Here are some suggestions that might improve the manuscript.

* A statistics section is missing. As this is a brief report, it could be ok but statisics need to be reported in e.g., figure or table legends

* Data in Figure 1 represent repeated measurements of the same patients. I suggest that the authors present individual data (e.g., dots and/or lines) and not only a box plot. I think some statitics with repeated measurements (e.g., paired T-test or linear mixed model) could improve this part of the manuscript.

* Some of the statements in the conclusion are not supported by the results due to the fact that this is a single-centre experience with a low number of patients. I think the conclusion could be shortend and focused (see abstract).

Author Response

Point 1: * A statistics section is missing. As this is a brief report, it could be ok but statisics need to be reported in e.g., figure or table legends

Response 1: Thank you! I will add a brief statistics sentence in the "Methods" section that we used chi-squared test and paired t-test.  For figure 1, I will add the statistics in the caption section.  

Point 2: Data in Figure 1 represent repeated measurements of the same patients. I suggest that the authors present individual data (e.g., dots and/or lines) and not only a box plot. I think some statitics with repeated measurements (e.g., paired T-test or linear mixed model) could improve this part of the manuscript

Response 2: A dots and lines graph had been attempted previously, but the diagram was too confusing to track each individual patient's hemoglobin on the y-axis over time on the x-axis, losing sight of the dots and lines.  It was simplified to a box and whiskers plot to highlight the hemoglobin drop and recovery across all the patients.  I will add a paired t-test statistic in the caption section of figure 1 to compare the baseline hemoglobin to nadir and the nadir to hemoglobin recovery after dapsone discontinuation.   Baseline to nadir p=0.0018 and nadir to hemoglobin recovery p < 0.0001

Point 3:  Some of the statements in the conclusion are not supported by the results due to the fact that this is a single-centre experience with a low number of patients. I think the conclusion could be shortend and focused (see abstract)

Response 3: I will shorten the conclusion section and highlight the need for closer monitoring of patients with normal G6PD function on dapsone prophylaxis

Round 2

Reviewer 2 Report

Corrections are performed and questions are adequately answered.